# An Entropy-Based Cross-Efficiency under Variable Returns to Scale

**DOI:** 10.3390/e21121205

**Published:** 2019-12-07

**Authors:** Chun-Hsiung Su, Tim Lu

**Affiliations:** 1Department of Tourism and Leisure Management, Vanung University, Zhongli, Taoyuan 320, Taiwan; samsu@mail.vnu.edu.tw; 2Department of Marketing and Logistics, Vanung University, Zhongli, Taoyuan 320, Taiwan

**Keywords:** data envelopment analysis, cross efficiency, Shannon’s entropy, variable returns to scale, ranking

## Abstract

Cross-efficiency evaluation is an effective methodology for discriminating among a set of decision-making units (DMUs) through both self- and peer-evaluation methods. This evaluation technique is usually used for data envelopment analysis (DEA) models with constant returns to scale due to the fact that negative efficiencies never happen in this case. For cases of variable returns to scale (VRSs), the evaluation may generate negative cross-efficiencies. However, when the production technology is known to be VRS, a VRS model must be used. In this case, negative efficiencies may occur. Negative efficiencies are unreasonable and cause difficulties in calculating the final cross-efficiency. In this paper, we propose a cross-efficiency evaluation method, with the technology of VRS. The cross-efficiency intervals of DMUs were derived from the associated aggressive and benevolent formulations. More importantly, the proposed approach does not produce negative efficiencies. For comparison of DMUs with their cross-efficiency intervals, a numerical index is required. Since the concept of entropy is an effective tool to measure the uncertainty, this concept was employed to build an index for ranking DMUs with cross efficiency intervals. A real-case example was used to illustrate the approach proposed in this paper.

## 1. Introduction

Data envelopment analysis (DEA) is a non-parametric method for efficiency evaluation of a group of homogeneous decision-making units (DMUs) that consume multiple inputs to produce multiple outputs. Since DEA is a non-parametric method, it does not require any predetermined information on the production function of the production entities before evaluation. That is, the evaluation results are obtained from the input and output data and obtained by comparing the production of each DMU with those of the others. For its effectiveness in identifying the best-practice frontier and ranking the DMUs, DEA has been widely applied in many different sectors and industries. However, traditional self-evaluated DEA models with total weight flexibility may evaluate many DMUs as DEA-efficient and cannot make any further distinctions among them. Therefore, one of the main shortfalls of the traditional DEA models is their inability to discriminate among DMUs that are all deemed efficient [1].

While allowing every DMU to select different multipliers to measure efficiency is a merit of the DEA technique, this makes the resulting efficiencies of the DMUs incomparable. The efficiencies are comparable only if they are calculated from the same set of weights. In cross-efficiency evaluation, each DMU defines its most favorable weights associated with the inputs and outputs for self-efficiency evaluation. Using these weights, it can also evaluate the efficiencies of the other DMUs, which gives rise to peer-evaluated efficiencies. For each DMU under evaluation, we can obtain a final efficiency by aggregating its self-evaluated efficiency and its efficiencies peer-evaluated by the others. In this case, every DMU has *n* efficiency scores calculated from *n* sets of weights selected by all *n* DMUs, including itself. The average of the *n* efficiency scores is the final efficiency for this DMU.

Cross-efficiency evaluation almost always ranks the DMUs in a unique order [2] and eliminates unrealistic weight schemes without incorporating weight restrictions [3]. Due to these advantages, cross-efficiency evaluation has been extensively applied in performance evaluation. Since the seminal work of Sexton et al. [4] and Doyle and Green [2], a number of cross-efficiency models and applications have been reported in the literature. Under constant returns to scale (CRS), Liang et al. [5] proposed a game cross-efficiency model to generate a set of cross efficiencies that constitutes a Nash equilibrium point for the DMUs. Jahanshahloo et al. [6] incorporated a symmetric technique into the cross-efficiency evaluation that could choose symmetric weights for DMUs. There are some methods that select suitable weights from alternative solutions to avoid large differences among the weights. Setting lower bounds [7,8], using ordered weighted averaging operators [9], and evaluating the robustness of the proposed methodology [10] are some examples. Wu et al. [11] developed a target identification model to obtain reachable targets for all DMUs, and several secondary goals were proposed considering both the desirable and undesirable targets.

Cross-efficiency evaluation is usually used for production technologies with CRS because negative efficiencies never happen in this case. However, when the production technology is known to be a variable return to scale (VRS), a VRS model must be used. In this case, negative efficiencies may occur. Negative efficiencies are unreasonable and cause difficulties in calculating the final cross-efficiency. In the literature, only a couple of studies calculate cross efficiencies under the VRS technology. Wu et al. [12] and Soares de Mello et al. [13] proposed the idea of restricting the multiplier values to those that could only produce positive efficiencies for all DMUs. Lim and Zhu [14] translated the coordinates to let negative efficiencies become positive. Lin [15] adopted the range-directional measure proposed in Portela et al. [16] to calculate efficiencies.

Existing approaches for cross-efficiency evaluations are often averaging the entries of the cross-efficiency matrix column-wise, that is, the average cross-efficiency, to further discriminate among the DEA efficient units. In this case, the problem of choosing the aggressive (lower bound efficiency) or benevolent formulation (upper bound efficiency) for decision-making might still remain. In this paper, we propose a cross-efficiency evaluation method, with the technology of VRS. The cross-efficiency of a DMU is calculated as an interval, where the lower bound and upper bound are obtained by aggressive and benevolent formulations, respectively. In other words, the cross-efficiency interval takes the aggressive and benevolent formulations into account at the same time, and the choice of aggressive/benevolent formulation can be avoided. More importantly, the proposed approach does not produce negative efficiencies. For comparison of DMUs with their cross-efficiency intervals, a numerical index is required. Since the concept of entropy [17] is an effective tool to measure the uncertainty, this concept is employed to construct an index for ranking DMUs with cross efficiency intervals.

In the sections that follow, we first introduce the cross-efficiency evaluation under the assumption of VRS in Section 2. Then, Section 3 introduces the concept of Shannon’s entropy and develops a solution procedure to find the optimal entropy value for comparison of DMUs. A real-case example is used to illustrate the approach proposed in this study in Section 4. Finally, some conclusions of this study are presented in Section 5.

## 2. Negative Cross-Efficiency

In this section, we illustrate the problem of negative cross-efficiency. Suppose that we have *n* DMUs, where every DMU *j*, *j* = 1, …, *n*, produces the same *s* outputs in different amounts, *Y_rj_* (*r* = 1, …, *s*), using the same *m* inputs, *X_ij_* (*i* = 1, …, *m*) in different amounts. The VRS model [18], which was developed by Banker, Chranes, and Cooper (BCC), for measuring the efficiency of DMU *d* under variable returns to-scale, has output- and input-oriented models. The output-oriented VRS model is formulated as:(1)$$\begin{array}{c} 1/E_{dd}^{O}=\min\;\sum_{i=1}^{m}v_{id}X_{id}+v_{0d}\\ s.t.\;\sum_{r=1}^{s}\;u_{rd}\;Y_{rd}\;=\;1\\ \sum_{r=1}^{m}v_{id}X_{ij}+v_{0d}-\sum_{r=1}^{s}u_{rd}Y_{rj}\geq 0,j=1,\ldots ,n\\ u_{rd},\;v_{id}\;\geq \;0,\;r\;=\;1,\;\ldots ,\;s,\;i\;=\;1,\;\ldots ,\;m\\ v_{0d}\;\mathrm{unrestricted}\;\mathrm{in}\;\mathrm{sign}, \end{array}$$
where $u_{rd}$ and $v_{id}$ are the multipliers selected by DMU *d* to calculate efficiencies. A self-evaluated efficiency score of DMU *d* and the optimal weights are obtained from solving Model (1). For the special cases of $v_{0d}$ = 0 in (1), the model becomes the CCR model [19], with a technology of CRS, and the derived efficiency score is regarded as the CCR efficiency. In a VRS model, the variable $v_{0d}$ gives an indication of the type of returns to scale that prevails at a particular DMU under evaluation. Especially, when the optimal solution in (1) $v_{0d}^{*}$ < 0 (>0), it indicates that the DMU under evaluation is the increasing (decreasing) returns to scale.

Specifically, if $v_{id}^{*}$ (*i* = 1, …, *m*) and $u_{rd}^{*}$ (*r* = 1, …, *s*) is an optimal solution of (1) for a given DMU *d*, then a cross-efficiency of DMU *j* peer-evaluated by DMU *d* is given by
(2)$$1/E_{dj}^{O}=\frac{\sum_{i=1}^{m}v_{id}^{*}X_{ij}+v_{0d}}{\sum_{r=1}^{s}u_{rd}^{*}Y_{rj}},\;d,\;j=\;1,\;\ldots ,\;n.$$

Since $\sum_{r=1}^{s}u_{rd}Y_{rj}>0$ in the second constraint of Model (1), the term $\sum_{i=1}^{m}v_{id}X_{ij}+v_{0d}$ must be greater than 0. That implies that the cross-efficiency $1/E_{dj}^{O}$ > 0. In other words, the problem of negative cross-efficiency will not occur in the output-oriented VRS model.

On one hand, the input-oriented VRS model is formulated as:(3)$$\begin{array}{c} E_{dd}^{I}\;=\;\max\;\sum_{r=1}^{s}u_{rd}Y_{rd}+u_{0d}\\ s.t.\;\sum_{i=1}^{m}v_{id}X_{id}\;=\;1\\ \sum_{r=1}^{s}u_{rd}Y_{rj}+u_{0d}-\sum_{i=1}^{m}v_{id}X_{ij}\leq 0,\;j=\;1,\;\ldots ,\;n,\\ u_{rd},\;v_{id}\geq 0,\;r=\;1,\;\ldots ,\;s,\;i=\;1,\;\ldots ,\;m,\\ u_{0d}\;\mathrm{unrestricted}\;\mathrm{in}\;\mathrm{sign}. \end{array}$$

Similarly, let $v_{id}^{*}$ (*i* = 1, …, *m*), $u_{rd}^{*}$ (*r* = 1, …, *s*), and $u_{0d}^{*}$ be an optimal solution of (3) for a given DMU *d*; then, the cross-efficiency is given by
(4)$$E_{dj}^{I}=\frac{\sum_{r=1}^{s}u_{rd}^{*}Y_{rj}+u_{0d}^{*}}{\sum_{i=1}^{m}v_{id}^{*}X_{ij}},\;d,\;j\;=\;1,\;\ldots ,\;n.$$

We can repeat this process and use the weights selected by every DMU for calculating the efficiencies of all DMUs. The final cross-efficiency of DMU *j* is the average of $E_{dj}$, *d* = 1, …, *n*; that is,
(5)$${\bar{E}}_{j}^{I}=\frac{1}{n}\sum_{d=1}^{n}E_{dj},\;j=1,\ldots ,n$$

The cross-efficiency score ${\bar{E}}_{j}^{I}$ provides a peer-evaluation of DMU *j*, and the derived values can, thus, be used for ranking DMUs. However, due to the unrestricted variable $u_{0d}^{*}$, the value of $\sum_{r=1}^{s}u_{rd}^{*}Y_{rj}+u_{0d}^{*}$ could be negative when $u_{0d}^{*}$ < 0, and the cross-efficiency calculated by Equation (4) may lead to a problematic situation. Negative efficiencies are obviously unreasonable, and we need to develop a procedure to tackle this problem. Note that negative efficiencies will not happen in the CCR model, nor in the VRS output-oriented model.

In the next section, we propose a methodology to calculate and rank cross-efficiencies for the input-oriented VRS model.

## 3. Entropy with VRS Cross-Efficiencies

### 3.1. VRS Cross-Efficiencies

The most commonly used secondary goals approach is proposed by Doyle and Green [2]. They defined the aggregate efficiency to be the weighted average of the other *n*−1 efficiencies, with the weight of $\sum_{i=1}^{m}v_{id}X_{id}$/$\sum_{j=1,j\neq d}^{n}\sum_{i=1}^{m}v_{id}X_{ij}$ for DMU *d* to obtain the following model:(6)$$\begin{array}{c} \max\;\frac{\sum_{j=1,j\neq d}^{n}\sum_{r=1}^{s}u_{rd}Y_{rd}}{\sum_{j=1,j\neq d}^{n}\sum_{i=1}^{m}v_{id}X_{id}}\\ s.t.\;\sum_{r=1}^{s}u_{rd}Y_{rd}\;=\;E_{dd}\sum_{i=1}^{m}v_{id}X_{ij}\\ \sum_{r=1}^{s}u_{rd}Y_{rj}-\sum_{i=1}^{m}v_{id}X_{ij}\leq 0,\;j\;=\;1,\;\ldots ,\;n\\ u_{rd},\;v_{id}\;\geq \;0,\;r\;=\;1,\;\ldots ,\;s,\;i\;=\;1,\;\ldots ,\;m \end{array}$$
where $E_{dd}$ is the CCR efficiency of DMU *d*. This model is a linear fractional program, which can be linearized by applying the variable substitution technique of Charnes and Cooper [20] as follows:(7)$$\begin{array}{c} \max\sum_{j=1,j\neq d}^{n}\sum_{r=1}^{s}u_{rd}Y_{rd}+u_{0d}\\ s.t.\;\sum_{j=1,j\neq d}^{n}\sum_{i=1}^{m}v_{id}X_{id}\;=\;1\\ \sum_{r=1}^{s}u_{rd}Y_{rd}\;=\;E_{dd}\sum_{i=1}^{m}v_{id}X_{ij}\\ \sum_{r=1}^{s}u_{rd}Y_{rj}-\sum_{i=1}^{m}v_{id}X_{ij}\leq \;0,\;j\;=\;1,\;\ldots ,\;n\\ u_{rd},\;v_{id}\;\geq \;0,\;r\;=\;1,\;\ldots ,\;s,\;i\;=\;1,\;\ldots ,\;m \end{array}$$

Since this model is to search the maximum cross-efficiency, Doyle and Green [2] named this model the benevolent model. On the other hand, when the objective function in Model (7) is changed to the minimum operation, the formulation becomes an aggressive model. Model (7) is under the assumption of CRS, and we can rewrite Model (7) to the input-oriented VRS model. Let $E_{dd}^{A}$ and $E_{dd}^{B}$ be the efficiency scores of aggressive and benevolent models under the assumption of VRS, respectively, and they can be expressed by:(8)$$\begin{array}{c} E_{dd}^{A}=\min\sum_{j=1,j\neq d}^{n}\sum_{r=1}^{s}u_{rd}Y_{rd}+u_{0d}\\ s.t.\;\sum_{j=1,j\neq d}^{n}\sum_{i=1}^{m}v_{id}X_{id}\;=\;1\\ \sum_{r=1}^{s}u_{rd}Y_{rd}+u_{0d}\;=\;E_{dd}\sum_{i=1}^{m}v_{id}X_{ij}\\ \sum_{r=1}^{s}u_{rd}Y_{rj}+u_{0d}-\sum_{i=1}^{m}v_{id}X_{ij}\leq \;0,\;j\;=\;1,\;\ldots ,\;n\\ u_{rd},\;v_{id}\geq \;0,\;r\;=\;1,\;\ldots ,\;s,\;i\;=\;1,\;\ldots ,\;m\\ u_{0d}\;\mathrm{unrestricted}\;\mathrm{in}\;\mathrm{sign}. \end{array}$$
(9)$$\begin{array}{c} E_{dd}^{B}=\max\sum_{j=1,j\neq d}^{n}\sum_{r=1}^{s}u_{rd}Y_{rd}+u_{0d}\\ s.t.\;\sum_{j=1,j\neq d}^{n}\sum_{i=1}^{m}v_{id}X_{id}\;=\;1\\ \sum_{r=1}^{s}u_{rd}Y_{rd}+u_{0d}\;=\;E_{dd}\sum_{i=1}^{m}v_{id}X_{ij}\\ \sum_{r=1}^{s}u_{rd}Y_{rj}+u_{0d}-\sum_{i=1}^{m}v_{id}X_{ij}\leq \;0,\;j\;=\;1,\;\ldots ,\;n\\ u_{rd},\;v_{id}\;\geq \;0,\;r\;=\;1,\;\ldots ,\;s,\;i\;=\;1,\;\ldots ,\;m\\ u_{0d}\;\mathrm{unrestricted}\;\mathrm{in}\;\mathrm{sign}. \end{array}$$

Since the cross-efficiency is calculated by Equation (4), the problem of the negative efficiency arises when $\sum_{r=1}^{s}u_{rd}^{*}Y_{rj}+u_{0d}^{*}$ < 0. To avoid the occurrence of the negative efficiency, Wu et al. [12] and Soares de Mello et al. [13] proposed to add constraints $\sum_{r=1}^{s}u_{rd}Y_{rj}+u_{0d}\geq 0$, *j* = 1, …, *n*, to Equation (3). Following this idea, we add this constraint to Models (8) and (9), and they become:(10)$$\begin{array}{c} E_{dd}^{A}=\min\sum_{j=1,j\neq d}^{n}\sum_{r=1}^{s}u_{rd}Y_{rj}+u_{0d}\\ s.t.\;\sum_{j=1,j\neq d}^{n}\sum_{i=1}^{m}v_{id}X_{id}\;=\;1\\ \sum_{r=1}^{s}u_{rd}Y_{rd}+u_{0d}\;=\;E_{dd}\sum_{i=1}^{m}v_{id}X_{ij}\\ \sum_{r=1}^{s}u_{rd}Y_{rj}+u_{0d}-\sum_{i=1}^{m}v_{id}X_{ij}\leq \;0,\;j\;=\;1,\;\ldots ,\;n\\ \sum_{r=1}^{s}u_{rd}Y_{rj}+u_{0d}\geq 0,\;j\;=\;1,\ldots ,\;n\\ u_{rd},\;v_{id}\;\geq \;0,\;r\;=\;1,\;\ldots ,\;s,\;i\;=\;1,\;\ldots ,\;m\\ u_{0d}\;\mathrm{unrestricted}\;\mathrm{in}\;\mathrm{sign}. \end{array}$$
(11)$$\begin{array}{c} E_{dd}^{B}=\max\sum_{j=1,j\neq d}^{n}\sum_{r=1}^{s}u_{rd}Y_{rj}+u_{0d}\\ s.t.\;\sum_{j=1,j\neq d}^{n}\sum_{i=1}^{m}v_{id}X_{id}\;=\;1\\ \sum_{r=1}^{s}u_{rd}Y_{rd}+u_{0d}\;=\;E_{dd}\sum_{i=1}^{m}v_{id}X_{ij}\\ \sum_{r=1}^{s}u_{rd}Y_{rj}+u_{0d}-\sum_{i=1}^{m}v_{id}X_{ij}\leq \;0,\;j\;=\;1,\;\ldots ,\;n\\ \sum_{r=1}^{s}u_{rd}Y_{rj}+u_{0d}\geq 0,\;j\;=\;1,\ldots ,\;n\\ u_{rd},\;v_{id}\;\geq \;0,\;r\;=\;1,\;\ldots ,\;s,\;i\;=\;1,\;\ldots ,\;m\\ u_{0d}\;\mathrm{unrestricted}\;\mathrm{in}\;\mathrm{sign}. \end{array}$$

In Models (10) and (11), the second constraint $\sum_{r=1}^{s}u_{rd}Y_{rd}+u_{0d}$ = $E_{dd}\sum_{i=1}^{m}v_{id}X_{ij}$ is the secondary goal, which is used to deal with the multiple optimum weights and produce the same efficiency $E_{dd}$ for DMU *d*. We can find $v_{id}^{*}$, $u_{rd}^{*}$, and $u_{0d}^{*}$ from solving Model (10), according to the studies of Wu et al. [12] and Soares de Mello et al. [13], the values of $v_{id}^{*}$, $u_{rd}^{*}$, and $u_{0d}^{*}$ of DMU *d* are put into Equation (4) to obtain the cross-efficiencies $E_{dj}^{I}$; that is, $E_{dj}^{I}=\left(\sum_{r=1}^{s}u_{rd}^{*}Y_{rj}+u_{0d}^{*}\right)/\sum_{i=1}^{m}v_{id}^{*}X_{ij}$. Since $\sum_{r=1}^{s}u_{rd}^{*}Y_{rj}^{*}+u_{0d}^{*}\geq 0$ and $\sum_{r=1}^{s}v_{id}^{*}X_{ij}^{*}>0$, we have $E_{dj}^{I}$ > 0. In other words, the approach proposed in this paper does not produce negative efficiencies. If there are n DMUs, then we have n different sets of $v_{id}^{*}$, $u_{rd}^{*}$, and $u_{0d}^{*}$. By putting the obtained n different sets of $v_{id}^{*}$, $u_{rd}^{*}$, and $u_{0d}^{*}$ into Equation (4), we have n different value of $E_{dj}^{I}$ to find the final cross-efficiency E¯jI=1n∑d=1nEdj, j=1,…,n. With the same process, we can derive the final cross-efficiency for Model (11) as well.

### 3.2. The Entropy

The idea of Shannon’s entropy plays a central role in information theory. Based on Ormos and Zibriczky [21], entropy is a mathematically defined quantity, usually applied to describe the probability of results in a system. Since Shannon’s entropy provides a powerful tool for the measurement of uncertainty, this concept has been applied in many scientific fields, such as mechanics, statistics, transport, information theory, and mathematical programming problems. In the literature, several articles applied the entropy concept and DEA models for ranking DMUs. Soleimani-Damaneh and Zarepisheh [22] applied Shannon’s entropy to integrate a family of DEA efficiencies, which are calculated from different DEA models, into an index for distinguishing DMUs. The Shannon’s entropy was used by Xie et al. [23] to merge the efficiency scores and help discriminate traditional DEA models. Wang et al. [24] explored the DEA entropy model to construct the cross-efficiency intervals for ranking DMUs. Lu and Liu [25] considered the benevolent and aggressive and formulations simultaneously for obtaining a number of cross-efficiency intervals, and the entropy was used to build a numerical index for the DMUs to be comparable. Rotela Junior et al. [26] adopted a DEA model in portfolio optimization, and Shannon entropy was included to ensure an efficient asset diversification while return and portfolio risk were maximized and minimized, respectively. Lee [27] applied the concept of Shannon’s entropy to combine cross-efficiency scores, which are measured from different cross-efficiency evaluation models, for discrimination of DMUs. According to the relative entropy and grey relational analysis methods, Si and Ma [28] proposed a cross-efficiency method for a comparison of DMUs. However, these studies which measured the cross-efficiency scores were all under the assumption of CRS.

In this section, we employ Shannon’s entropy to integrate the derived cross-efficiency intervals for distinguishing DMUs. Since we take into account the aggressive and benevolent cross-efficiency at the same time, a family of cross-efficiency intervals is derived for each DMU. The idea of Lu and Liu [25], which calculated the entropy value of the cross-efficiency interval, is adopted to construct a numerical index for DMUs with cross efficiency intervals.

If the cross-efficiency of DMU *j* is a constant value, then the entropy for DMU *j* can be defined as:(12)$$H_{j}=-K_{j}\sum_{d=1}^{n}\left(\frac{E_{dj}}{\sum_{d=1}^{n}E_{dj}}\ln\frac{E_{dj}}{\sum_{d=1}^{n}E_{dj}}\right),$$
where $K_{j}$ is a constant value. In this study we derive the aggressive and benevolent cross-efficiencies $E_{dj}^{A}$ and $E_{dj}^{B}$ from Models (10) and (11), respectively. That is, the cross-efficiency is an interval rather than a constant. In this case, we should rewrite Equation (12) to the following form:(13)$${\hat{H}}_{j}=-K_{j}\sum_{d=1}^{n}\left(\frac{{\hat{E}}_{dj}}{\sum_{d=1}^{n}{\hat{E}}_{dj}}\ln\frac{{\hat{E}}_{dj}}{\sum_{d=1}^{n}{\hat{E}}_{dj}}\right),$$
where ${\hat{E}}_{dj}\in \left[E_{dj}^{A},\;E_{dj}^{B}\right]$.

To find the smallest uncertainty of a DMU with $E_{dj}^{A}$ and $E_{dj}^{B}$, we should find the minimum value of entropy represented in (13). With the smallest entropy (uncertainty) of each DMU, we can discriminate DMUs more easily. Based on Lu and Liu [25], Equation (13) can be transformed into
(14)$${\hat{H}}_{j}\;=\;\min\;-K_{j}\sum_{d=1}^{n}\left(\frac{{\hat{E}}_{dj}}{\sum_{d=1}^{n}{\hat{E}}_{dj}}\ln\frac{{\hat{E}}_{dj}}{\sum_{d=1}^{n}{\hat{E}}_{dj}}\right)$$
(15)$$s.t.\;E_{dj}^{A}\leq {\hat{E}}_{dj}\leq E_{dj}^{B},\;d=1,\ldots ,n.$$

We also follow the idea of Lu and Liu [25] to let the constant $K_{j}=\left({\bar{E}}_{j}^{A}+{\bar{E}}_{j}^{B}\right)/2$ in (14), where ${\bar{E}}_{j}^{A}$ and ${\bar{E}}_{j}^{B}$ are the final aggressive and benevolent cross-efficiencies that are calculated from Equation (5), respectively. According to Charnes and Cooper’s rule [20], we let $t=1/\sum_{d=1}^{n}{\hat{E}}_{dj}$ and $w_{dj}=t{\hat{E}}_{dj}$ by multiplying constraint (15) with *t* and replacing Model (14) with the following mathematical program:(16)$$\begin{array}{c} {\hat{H}}_{j}=\min\;-K_{j}\sum_{d=1}^{n}w_{dj}\ln\;w_{dj}\\ s.t.\;\sum_{d=1}^{n}w_{dj}\;=\;1,\\ E_{dj}^{L}t\leq w_{dj}\leq E_{dj}^{U}t,\;d=1,\ldots ,n.\\ t>0. \end{array}$$

Lu and Liu indicated that Model (16) is a concave function subject to linear constraints, and we can obtain the global optimum solution for (16). We can discriminate all DMUs with the derived value of ${\hat{H}}_{j}^{*}$, and the larger the value of ${\hat{H}}_{j}^{*}$ the better the DMU is.

## 4. Example

Lee [27] studied the problem of calculating the cross-efficiency scores for commercial banks in Taiwan. Three inputs and three outputs are considered when measuring the efficiencies, and the associated three input and three output items are listed as follows:

Inputs:

Labor cost.

Physical capital (book value of fixed assets for business purposes).

Purchase funds (including time and saving deposits, and other bank deposits).

Outputs:

Demand deposits (including checking, passbook, and temporary deposits).

Short-term loans.

Medium-and-long-term loans.

Table 1 lists the associated input and output data of the twenty-two bank. The dataset used by Lee [27] can demonstrate that our proposed approach is effective at distinguishing the efficient DMUs for complex problems. In this section we use this data set to explain how our proposed approach is applied to calculate the VRS cross-efficiencies and the entropy values of the commercial banks.

By applying the idea set forth in this paper, we first used the BCC model (7) to measure the efficiency for every commercial bank. The results are shown in the second column of Table 1 under the heading “BCC.” It is noted that twelve commercial banks are efficient, and their ranks cannot be differentiated. We then used Models (10) and (11) to calculate the aggressive (lower bound) and benevolent (upper bound) efficiencies of every commercial bank, respectively, with the results shown in Table 2. No negative values appear in the calculated results. The final efficiencies, as calculated from Equation (5), are shown in the last column of Table 2.

With the derived cross-efficiency intervals, we then applied Model (16) to calculate the corresponding entropy for commercial banks. The entropy cross-efficiency values and their associated ranks of banks are reported in Table 3. Since the higher the entropy cross-efficiency the more efficient the bank is, Bank number 3 is in first place, followed by Banks 16, 4, and 15 subsequently. This indicates that the approach proposed in this paper works well for complex problems for discriminating efficient DMUs.

## 5. Conclusions

Cross efficiency is an aggregate efficiency measured from the viewpoints of all DMUs. The results are, thus, more representative and persuasive than those of its counterparts. However, most studies are restricted to production technologies with CRS due to the possibility of producing negative efficiencies under VRS. In this paper, we proposed a cross-efficiency evaluation method, with the technology of VRS. Each DMU has a cross-efficiency interval, where aggressive and benevolent formulations are derived from, respectively. Since the concept of entropy is an effective tool to measure the uncertainty, this concept is used to construct an index for ranking DMUs with cross efficiency intervals.

The most important merit of the proposed approach is that this model does not produce negative efficiencies, which makes it appropriate for cases of VRS DEA models. A real-world case shows that the final efficiencies calculated from the cross efficiencies help identify the rankings of a set of DMUs. Cross-efficiency evaluation has been extended to different evaluation models. In this study, the input and output data were measured by exact values. However, in some cases, the inputs and outputs of DMUs may be stochastic. The derivation of the stochastic measure and its applications will be another direction for future studies.

## Figures and Tables

**Table 1 entropy-21-01205-t001:** Real data (in millions of Taiwanese dollars) and BCC efficiencies of 22 Taiwanese commercial banks in 2013.

Bank	Labor	Capital	Purchased Funds	Deposits	S-Term Loans	ML-Term Loans	BCC
1	9492	23,935	1,029,108	336,735	297,352	844,783	0.9363
2	848	2683	121,212	24,362	27,961	79,582	1.0000
3	2351	3416	323,449	106,247	104,348	259,497	1.0000
4	7306	14,299	815,246	279,769	339,261	617,217	1.0000
5	1388	2744	162,563	23,395	69,956	108,206	1.0000
6	1999	6195	125,917	15,016	30,227	69,487	0.9803
7	2838	7644	307,145	56,564	71,591	158,042	0.6872
8	3545	2814	325,073	48,824	48,539	247,323	0.9416
9	3585	3343	280,959	56,041	64,251	202,585	0.8905
10	1775	1128	204,472	21,517	36,705	150,177	1.0000
11	10,717	28,674	1,226,897	508,605	384,511	1,023,549	1.0000
12	9308	11,294	1,078,604	250,407	310,403	783,664	0.8749
13	9346	22,617	1,271,363	336,838	289,442	744,008	0.7941
14	6455	18,487	841,496	305,603	187,843	643,889	1.0000
15	3074	2150	395,750	66,537	92,533	307,930	1.0000
16	12,502	14,519	1,347,592	580,389	462,928	1,188,269	1.0000
17	9277	17,464	715,304	163,804	158,695	547,688	0.7989
18	3642	6915	505,286	105,395	103,643	341,020	0.8566
19	8049	11,002	616,242	232,732	218,083	594,174	1.0000
20	20,295	34,229	997,936	146,904	348,395	941,957	1.0000
21	11,405	27,730	1,243,848	476,748	404,671	1,028,704	0.9464
22	14,354	38,694	1,825,537	442,195	376,648	1,531,299	1.0000

**Table 2 entropy-21-01205-t002:** Lower bound and upper bound variable return to scale (VRS) cross-efficiencies of commercial banks in Taiwan.

**Bank**	**1**		**2**		**3**		**4**		**5**		**6**		**7**		**8**		**9**		**10**		**11**	
	**L**	**U**	**L**	**U**	**L**	**U**	**L**	**U**	**L**	**U**	**L**	**U**	**L**	**U**	**L**	**U**	**L**	**U**	**L**	**U**	**L**	**U**
1	0.9363	0.9363	0.9105	0.9119	1.0000	1.0000	0.8746	0.8746	0.8508	0.8520	0.6167	0.6191	0.6237	0.6244	0.8579	0.8587	0.7828	0.7840	0.9079	0.9088	0.9621	0.9623
2	0.0894	0.9136	1.0000	1.0000	0.3610	1.0000	0.1161	0.8573	0.6112	0.9120	0.4245	0.7190	0.2990	0.6499	0.2394	0.8770	0.2367	0.8162	0.4779	0.9468	0.0792	0.9337
3	0.5012	0.9136	0.3557	1.0000	1.0000	1.0000	0.8573	0.8945	0.8976	0.9120	0.1692	0.7190	0.3592	0.6499	0.4019	0.8770	0.5019	0.8162	0.6411	0.9468	0.5515	0.9337
4	0.5894	0.8132	0.0154	0.9311	0.8629	0.9573	1.0000	1.0000	0.7068	1.0000	0.0211	0.8238	0.3084	0.6731	0.1862	0.5197	0.3111	0.7018	0.1662	0.5862	0.6699	0.9464
5	0.6727	0.8955	0.5484	1.0000	0.9121	1.0000	0.9081	1.0000	1.0000	1.0000	0.2577	0.7603	0.5018	0.6698	0.2588	0.8207	0.3522	0.8077	0.3703	0.9051	0.7745	0.9221
6	0.3748	0.3758	1.0000	1.0000	0.6067	0.6075	0.5237	0.5249	0.9993	1.0000	0.9803	0.9803	0.5341	0.5346	0.4350	0.4355	0.5583	0.5588	0.6348	0.6351	0.3842	0.3852
7	0.8208	0.8246	1.0000	1.0000	0.9736	0.9747	0.9723	0.9742	1.0000	1.0000	0.8763	0.8773	0.6872	0.6872	0.5696	0.5808	0.7351	0.7403	0.6409	0.6514	0.9505	0.9505
8	0.7852	0.7856	0.7716	0.7727	0.9609	0.9610	0.7875	0.7877	0.7966	0.7973	0.4938	0.4950	0.5281	0.5287	0.9416	0.9416	0.8626	0.8629	1.0000	1.0000	0.7938	0.7942
9	0.7886	0.7890	0.9991	1.0000	1.0000	1.0000	0.8346	0.8390	0.9905	1.0000	0.7231	0.7253	0.6274	0.6290	0.8864	0.8885	0.8905	0.8905	1.0000	1.0000	0.8124	0.8143
10	0.0471	0.8614	0.4206	1.0000	0.3303	1.0000	0.0789	0.8292	0.4112	0.9299	0.1821	0.7088	0.1476	0.6334	0.4009	0.9131	0.3376	0.8559	1.0000	1.0000	0.0393	0.8744
11	0.7680	0.7809	0.1502	0.9606	0.8736	1.0000	0.8420	0.9239	0.3354	0.6323	0.0987	0.4504	0.3521	0.5632	0.1930	0.4347	0.2592	0.5038	0.0881	0.4768	1.0000	1.0000
12	0.5897	0.5945	0.3221	0.3239	0.9995	1.0000	1.0000	1.0000	0.9225	0.9261	0.1582	0.1583	0.3966	0.3979	0.3208	0.3251	0.4335	0.4338	0.4571	0.4593	0.6600	0.6651
13	0.8319	0.8860	0.0646	0.5700	0.8966	1.0000	0.8588	1.0000	0.5001	0.5326	0.0278	0.1916	0.3752	0.4639	0.2998	0.5707	0.3162	0.4783	0.2337	0.5953	0.9825	0.9992
14	0.8686	0.8760	0.2888	0.6178	0.9263	1.0000	0.8329	0.8516	0.3364	0.5356	0.0538	0.2147	0.3757	0.4731	0.4880	0.5618	0.3955	0.4821	0.4448	0.5869	1.0000	1.0000
15	0.2920	0.8408	0.0339	0.8761	0.5014	1.0000	0.3453	0.8251	0.1272	0.8585	0.0267	0.5675	0.1044	0.5803	0.5697	0.9248	0.3590	0.8473	0.6447	1.0000	0.3000	0.8545
16	0.3674	0.9011	0.0089	1.0000	0.7179	1.0000	0.6651	0.8703	0.3878	0.8690	0.0355	0.7239	0.1715	0.6490	0.2400	0.8035	0.3352	0.7884	0.1819	0.8565	0.4212	0.9640
17	0.8334	0.8507	0.9938	1.0000	0.9083	0.9289	0.7831	0.7993	0.9107	0.9144	0.8448	0.8795	0.6383	0.6425	0.8677	0.8710	0.8441	0.8505	0.9218	0.9297	0.8395	0.8596
18	0.8296	0.8298	0.7668	0.7692	1.0000	1.0000	0.7840	0.7840	0.6637	0.6649	0.2778	0.2790	0.4903	0.4910	0.6309	0.6310	0.5058	0.5061	0.7427	0.7430	0.8922	0.8923
19	0.8711	0.9011	0.0726	1.0000	0.7089	1.0000	0.8599	0.8703	0.3611	0.8690	0.0105	0.7239	0.3501	0.6490	0.5814	0.8035	0.5376	0.7884	0.4172	0.8565	0.9095	0.9640
20	0.8355	0.8441	0.0105	0.4589	0.6977	0.8253	1.0000	1.0000	0.5445	0.8324	0.0784	0.4280	0.3808	0.5330	0.3785	0.5367	0.4526	0.6247	0.2576	0.5173	0.8933	0.8984
21	0.8840	0.8845	0.3617	0.3763	0.9992	1.0000	1.0000	1.0000	0.6875	0.6957	0.1316	0.1394	0.4612	0.4655	0.4691	0.4739	0.4269	0.4322	0.4948	0.5020	0.9829	0.9836
22	0.8021	0.9350	0.1168	0.8760	0.7938	1.0000	0.7362	0.8730	0.2739	0.8205	0.0125	0.5556	0.3064	0.6067	0.4926	0.8342	0.3646	0.7490	0.4463	0.8854	0.8742	0.9655
Ave.	0.6536	0.8288	0.4642	0.8384	0.8196	0.9661	0.7573	0.8809	0.6507	0.8434	0.2955	0.5791	0.4100	0.5816	0.4868	0.7038	0.4909	0.6963	0.5532	0.7722	0.7169	0.8892
**Bank**	**12**		**13**		**14**		**15**		**16**		**17**		**18**		**19**		**20**		**21**		**22**	
	**L**	**U**	**L**	**U**	**L**	**U**	**L**	**U**	**L**	**U**	**L**	**U**	**L**	**U**	**L**	**U**	**L**	**U**	**L**	**U**	**L**	**U**
1	0.8436	0.8437	0.7055	0.7059	0.9163	0.9166	0.9488	0.9488	0.9999	1.0000	0.7965	0.7974	0.8333	0.8335	0.9990	1.0000	0.8195	0.8214	0.9422	0.9423	0.9877	0.9882
2	0.0912	0.8216	0.0908	0.6831	0.1314	0.8913	0.2760	0.9442	0.0679	0.9713	0.0915	0.7986	0.2330	0.8233	0.1054	1.0000	0.0418	0.8361	0.0744	0.9164	0.0591	0.9498
3	0.8216	0.8723	0.5106	0.6831	0.4177	0.8913	0.9442	1.0000	0.9713	1.0000	0.3316	0.7986	0.5370	0.8233	0.6542	1.0000	0.3661	0.8361	0.5865	0.9164	0.4009	0.9498
4	0.6888	0.8595	0.5933	0.6508	0.4808	0.7896	0.6186	0.6944	1.0000	1.0000	0.3365	0.6274	0.4943	0.6090	0.6417	0.9874	0.3952	0.6587	0.6988	0.9206	0.4872	0.5846
5	0.7168	0.8185	0.6646	0.6755	0.6164	0.8487	0.6130	0.9082	0.8018	0.9660	0.3598	0.7741	0.5844	0.7905	0.5769	1.0000	0.3699	0.8514	0.7666	0.9129	0.5662	0.8842
6	0.3695	0.3705	0.2973	0.2981	0.3306	0.3314	0.4665	0.4672	0.4069	0.4081	0.3489	0.3497	0.3870	0.3876	0.4997	0.5008	0.4368	0.4380	0.3949	0.3960	0.2540	0.2548
7	0.6861	0.6915	0.6527	0.6537	0.8125	0.8144	0.6448	0.6547	1.0000	1.0000	0.6479	0.6552	0.6314	0.6373	0.9966	1.0000	0.6544	0.6677	0.9201	0.9209	0.6073	0.6180
8	0.8316	0.8317	0.6035	0.6037	0.7492	0.7496	0.9997	1.0000	0.9966	0.9967	0.7303	0.7308	0.7591	0.7593	0.9997	1.0000	0.7897	0.7904	0.7978	0.7982	0.8159	0.8162
9	0.7858	0.7883	0.6077	0.6081	0.7529	0.7569	0.9319	0.9331	0.9593	0.9610	0.7246	0.7249	0.7477	0.7483	1.0000	1.0000	0.7856	0.7917	0.8126	0.8131	0.7535	0.7548
10	0.0999	0.8145	0.0499	0.6486	0.0610	0.8354	0.5249	0.9661	0.0777	0.9631	0.0646	0.7785	0.1632	0.8057	0.1025	1.0000	0.0330	0.8247	0.0407	0.8655	0.0292	0.8877
11	0.5862	0.6497	0.7136	0.7731	0.9110	0.9627	0.4333	0.5423	1.0000	1.0000	0.3679	0.4719	0.5585	0.6384	0.6203	0.7494	0.2251	0.2338	0.9024	0.9108	0.6217	0.6695
12	0.8749	0.8749	0.5953	0.5989	0.4976	0.5039	0.8440	0.8442	1.0000	1.0000	0.3594	0.3627	0.5694	0.5739	0.6679	0.6693	0.3963	0.3978	0.6887	0.6930	0.4803	0.4880
13	0.7960	0.7998	0.7941	0.7941	0.8666	1.0000	0.6309	0.8317	1.0000	1.0000	0.4484	0.5601	0.6582	0.8219	0.6752	0.7412	0.3927	0.4283	0.9336	0.9337	0.8349	1.0000
14	0.7761	0.7769	0.7852	0.7963	1.0000	1.0000	0.7550	0.8023	0.9972	1.0000	0.5199	0.5534	0.7746	0.8039	0.6975	0.7420	0.3872	0.4101	0.9262	0.9301	0.9632	1.0000
15	0.5701	0.8415	0.2689	0.6449	0.2801	0.8135	1.0000	1.0000	0.6947	1.0000	0.2468	0.7558	0.3540	0.8013	0.4299	0.9999	0.2298	0.7903	0.3118	0.8504	0.3406	0.8777
16	0.7609	0.7766	0.3800	0.6876	0.3144	0.9091	0.8568	0.8867	1.0000	1.0000	0.2401	0.7622	0.3513	0.7795	0.5573	1.0000	0.2607	0.7482	0.4470	0.9322	0.2963	0.8757
17	0.7406	0.7580	0.5983	0.6161	0.7892	0.8109	0.8603	0.8791	0.8820	0.9011	0.7989	0.7989	0.7369	0.7558	1.0000	1.0000	0.9224	0.9533	0.8320	0.8501	0.8321	0.8568
18	0.7842	0.7846	0.7407	0.7408	0.9260	0.9262	0.9135	0.9139	0.8894	0.8899	0.5467	0.5469	0.8566	0.8566	0.6848	0.6848	0.4334	0.4335	0.8427	0.8428	1.0000	1.0000
19	0.7766	0.7999	0.6303	0.6876	0.7670	0.9091	0.6937	0.8568	1.0000	1.0000	0.7460	0.7622	0.6119	0.7795	1.0000	1.0000	0.7482	1.0000	0.9111	0.9322	0.8757	0.8859
20	0.7889	0.7947	0.6248	0.6325	0.6743	0.7075	0.5816	0.6920	0.9666	0.9813	0.6538	0.6996	0.5271	0.6099	0.9641	1.0000	1.0000	1.0000	0.9083	0.9163	0.7381	0.7393
21	0.8718	0.8722	0.8132	0.8157	0.9047	0.9061	0.8290	0.8305	1.0000	1.0000	0.5264	0.5297	0.7938	0.7940	0.7256	0.7296	0.4730	0.4769	0.9464	0.9464	0.9651	0.9674
22	0.7534	0.8458	0.7087	0.7146	0.8738	0.9246	0.7617	0.9478	0.8787	1.0000	0.5062	0.7734	0.7321	0.8378	0.6401	0.9702	0.4221	0.7709	0.8259	0.9424	1.0000	1.0000
Ave.	0.6825	0.7858	0.5650	0.6688	0.6397	0.8272	0.7331	0.8429	0.8450	0.9563	0.4724	0.6642	0.5861	0.7396	0.6927	0.8988	0.4810	0.6891	0.7050	0.8674	0.6322	0.8204

**Table 3 entropy-21-01205-t003:** Entropy cross-efficiencies, and ranks for 22 commercial banks.

Bank	${\hat{H}}_{j}$	Rank
1	2.2119	10
2	1.7546	18
3	2.7260	1
4	2.4644	3
5	2.2349	8
6	1.1154	22
7	1.4895	21
8	1.7657	17
9	1.7826	16
10	1.9505	14
11	2.3925	5
12	2.2165	9
13	1.8491	15
14	2.1808	11
15	2.4034	4
16	2.7101	2
17	1.6881	20
18	2.0056	13
19	2.3890	6
20	1.7065	19
21	2.3479	7
22	2.1412	12

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
