# Peer review of "An Entropy-Based Cross-Efficiency under Variable Returns to Scale"

_entropy, 2019, doi:10.3390/e21121205_

Round 1

Reviewer 1 Report

REFEREE’S REPORT

This manuscript is entitled “An Entropy-Based Cross-Efficiency under Variable Return to Scale" for solving the negative efficiency scores under VRS with considering input-oriented. Although this topic is an interesting paper, however, several pieces of research have conducted different methods to measure and to rank, as the authors mentioned. In the first view, I felt the authors do not familiar with the basic concepts of DEA. This is because the efficiency scores cannot be less than zero. This is the ordinary sense for the DEA guys. After viewing the authors’ claim, I understand that the authors mixed and confused to use the "Indicator" and "Efficiency Score." Furthermore, the authors' reasoning can not convince me what the motivation has to do and why this is a critical issue. I just felt that all of the manipulations are mathematics formulas, such that there does not have enough practical implications. Besides, the illustrating study appears "comparison" but does not explain why the cases occur and why their model is suitable for these cases. Several concerns need the authors to clarify. Here, I list some critical issues, but not all.

The first and foremost, the authors mentioned that the BCC model with considering input-oriented might occur negative efficiency scores. They stated that the leading cause is the u0 term (i.e., unsigned) because they used the specific weights of input and output, and u0 to calculate the reference efficiency scores of other DMUs but excluded itself (peer review). And then average one –self-efficiency scores and the others -peer reviews. Conventionally, we would not adopt the u0 into the other DMUs because the return-to-scale has bundled into the weights of input and output while the specific DMU as the -self. The authors claimed that the u0 should take itself return-to-scale to others. This may confuse me that although the authors have adopted itself return-to-scale to others and forced the other DMUs may have the same return-to-scale with this specific DMU. However, this still cannot guarantee that the other DMUs will follow the same return-to-scale. Although the u0 may influence the specific DMU's efficiency scores by BCC, however, this u0 only just to us know what return-to-scale it is. More precisely, this information only helps us to know what positions and scale they are. The authors should clarify these issues.

Second, the purpose of Model 16 seems to find the degree of chaos rather than the efficiency scores. In information theory, entropy is the measure of the amount of information that is missing before reception and is sometimes referred to as Shannon entropy. The authors adopted the entropy to calculate efficiency scores that seem not suitable. This is also why the illustrating results can not comply with the BBC results. That is, some DMUs in the BCC are efficient, but ranking is lower in your proposed model. My viewpoint echoes this that the entropy cannot measure the efficiency scores. Many researchers have mentioned this concept. For example, the DMU 10 in the BCC model's efficiency score is one but cannot belong to the top 12 in your model (actually, this DMU's rank is 14 in the proposed model). Also, the DMU 12 is inefficient. However, the ranking is 9 in your model. The authors should clarify why these ranking does not make sense to our common sense. This showed that the cross-efficiency only could rank in the pooled but cannot apply to adjust resources.

Third, the Model 12 appears critical issues that the authors have "manipulated" the return to scale because the authors forced the sum (uY)+u0 >=0 in the third constraint; however, the u0 in BBC model determined which return-to-scale is for this specific DMU. That is, the u0 <0 represents to IRS, and the u0>0 represents to DRS. Based on previous comments, If the authors used a constraint sum(uY)+u0>=0 to lock the return-to-scale. The sum (uY) always more significant than zero; this implies the u0 is large than the zero. As a result, all the return-to-scale forced to be DRS. This does not make sense to me. The authors need to clarify.

Finally, the entire manuscript must be proofread first before any further submission.

Reviewer 2 Report

Minor revisions are needed. Some suggestions:

- DEA acronyms should be defined in the abstract.

- It should be explicit from the reading of the abstract what is the contribution and impact of this work. Currently, only the process is described but not the impact.

- The example given is interesting and pertinent. However, it is not clear the reasons for choosing this example. The choice should be based on objective and scientific criteria.

- The description of the inputs and outputs used in the example should be properly defined and explained. A good approach is to use a table to describe it.

- The date of the data shown in Table 1 is not indicated.

- Not all Table 3 values are aligned correctly.

- Indications should be given as to how this approach can be generalized. The existence of an example is useful, but not sufficient for generalization purposes.

- Conclusions should give some indications for future work.

Reviewer 3 Report

Introduction discusses earlier studies dealing with negative efficiency scores. However, there is no specific contribution of the present paper given in this regard.
Lines 73-77 should refer to specific numbering of sections rather than general words.
More recent literature on cross efficiency and entropy DEA should be discussed.
Why is model (12) needed? Please discuss.
Unclear term “mathematical form”, Line 177 contains “execute”
Line 193 is unclear: “demonstrates its useful disciplines”
Please provide interpretation of the H_j. Does this indicator measure the entropy or the level of efficiency? Entropy refers to uniform scores, whereas ranking should rely on the average levels. Thus, it is unclear what is measured by this indicator and whether it is relevant for ranking.
Line 228 contains a typo: “transform”
The proposed approach should be compared to the other cross-efficiency DEA model in Section 4.

Round 2

Reviewer 1 Report

First, I would like to thank the authors for quickly responding to my questions. However, I think the authors may confuse my concern. As Table A and Table B, my question is that authors used the common weights and common u0 to calculate the other DMUs, which implied that the authors had set all the DMUs that have the same u0 similar to this specific DMU, as P2 was shown. That is, if the u0 is negative, then all the DMUs have the same negative u0 while calculating the cross-efficiency scores. This demonstrated that all the DMUs are the IRS. This does not make sense to me. The return-to-scale for the model is not determined only by one data point. That is, the return-to-scale is forming by all DMUs. I understand that all the explanations by the authors. The authors find the u0 is calculated by the aggressive model and benevolent model and used the u0 to the other DMUs. My question is that the return-to-scale for this specific DMU has bundled to the common weights, while Model 10 and Model 11 were applied. More precisely, the common weights were based on the return-to-scale of this specific DMU. Why the authors used the u0 to calculate the other DMUs’ cross-efficiency scores were make sense? What the return to scale for the other DMUs?. The authors need to prove more evidences that they set all the other DMU has the same u0 is reasonable.

Reviewer 3 Report

Thank you for your revisions. the paper can be accepted for publication

Author Response

The opinion of this reviewer is that our paper can be accepted for publication. We appreciate the reviewer for his/her constructive comments for improving the quality of this paper. 

Round 3

Reviewer 1 Report

I want to thank the authors for replying to my concerns. However, the authors still do not proper response to my concerns on why the u0 used to other DMU’s to calculate their cross-efficiency scores reasonable? What the return to scale for the other DMUs if adopted one specific DMU’s u0? Instead, the authors mention that they followed by the idea of Wu et al. [12] and Sores de Mello et al. [13] to obtain the cross-efficiency scores, this may cause the negative efficiency scores. In my view is that the u0 has been“absorbed” and adequately represented to this specific DMU’s common weights; therefore, it might not need to put into the P2. If the authors set the common weights and common u0 into the P2, thus, all the DMUs have the same return-to-scale similar to this specific DMU. This case is also why the negative efficiency score occurs. Alternatively, if the u0 of this specific DMU does not put into the P2, then the negative cross-efficiency scores did not happen. The authors need to clarify this issue.

Author Response

Thanks for the reviewer's comment, and our response is as the attached file.
